# Obesity in Saudi Arabia in 2020: Prevalence, Distribution, and Its Current Association with Various Health Conditions

**DOI:** 10.3390/healthcare9030311

**Published:** 2021-03-11

**Authors:** Nora A. Althumiri, Mada H. Basyouni, Norah AlMousa, Mohammed F. AlJuwaysim, Rasha A. Almubark, Nasser F. BinDhim, Zaied Alkhamaali, Saleh A. Alqahtani

**Affiliations:** 1Sharik Association for Health Research, Riyadh 13326, Saudi Arabia; mada.basyouni@sharikhealth.net (M.H.B.); ralmubark@sharikhealth.net (R.A.A.); nasser.bindhim@sharikhealth.net (N.F.B.); 2Ministry of Health, Riyadh 11176, Saudi Arabia; 3Public Health Department, Imam Abdulrahman Bin Faisal University, Dammam 31441, Saudi Arabia; 2170001797@iau.edu.sa; 4Pharmacy College, King Faisal University, AlAhsa 31982, Saudi Arabia; 216006361@student.kfu.edu.sa; 5College of Medicine, Alfaisal University, Riyadh 11533, Saudi Arabia; 6Saudi Food and Drug Authority, Riyadh 13513, Saudi Arabia; ZKhamaali@sfda.gov.sa; 7Liver Transplant Unit, King Faisal Specialist Hospital & Research Centre, Riyadh 11211, Saudi Arabia; salqaht1@jhmi.edu; 8Division of Gastroenterology and Hepatology, Johns Hopkins University, Baltimore, MD 21218, USA

**Keywords:** Saudi Arabia, obesity, prevalence, noncommunicable diseases, body mass index

## Abstract

The global prevalence of obesity is increasing. Obesity is associated with many chronic diseases and health conditions. This study aims to estimate the current prevalence of obesity in Saudi Arabia and described the current national-level status of the association between obesity and various health conditions. This study is a nationwide cross-sectional survey conducted over phone-interviews in June 2020. In this study, a proportional quota-sampling technique was used to obtain equal distributions of participants, stratified by age and gender, across the 13 regions of Saudi Arabia. Weight and height were self-reported, and the obesity was determined as BMI ≥ 30. Logistic regression adjusted for age and gender was used for exploring current associations between obesity and health conditions. Of the 6239 participants contacted, 4709 participants responded and completed the interview with a response rate of 75.48%. Of them, 50.1% were female, the mean age was 36.4 ± 13.5 (Range: 18–90), and the median age was 36. The national weighted prevalence of obesity (BMI ≥ 30) was 24.7%, and the prevalence in the sample (unweighted) was 21.7%. Obesity was significantly associated with type 2 diabetes [Odd ratio, (OR) = 1.52], hypercholesterolemia (OR = 1.69), hypertension (OR = 1.61), lung diseases (OR = 1.69), rheumatoid arthritis (OR = 1.57), sleep apnea (OR = 1.82), colon diseases (OR = 1.31), and thyroid disorders (OR = 1.8). This study provides an update on the recent prevalence of obesity in Saudi Arabia. It also shows the variation in prevalence rates between different regions, which might be explored further. Although obesity shows a decreasing trend, almost one-quarter of this study sample were obese. Obesity is currently associated with many health conditions that can affect the individuals’ quality of life, impose stress on the healthcare system and impose an economic burden on the country. This evidence highlights the need for action to focus more on obesity in Saudi Arabia.

## 1. Introduction

Per the 2016 World Health Organization data, about 13% of the world’s adult population (male: 11%; female: 15%) were obese [1]. However, the Global Burden of Diseases (GBD) 2015 collaborators estimate that prevalence of obesity in adults in the Eastern Mediterranean Region (EMR) increased from 15% in 1980 to 21% in 2015, which is far higher than the global average of 12% in 2015 [2]. In 2017, a study conducted in the United Arab Emirates (UAE) has shown that the prevalence of overweight and obesity, were 43.0% and 32.3%, respectively [3]. In addition, in 2016 the obesity prevalence was estimated to be 31.2% in Bahrain, 26.1% in Israel, 28.3% in Oman, and 17.0% in Yemen [4]. A national survey conducted between 1995 and 2000 found that the overall prevalence of obesity among Saudi adults was 35.6% [5]. The national-level study in 2013 indicated a prevalence of 28.7% (men: 24.1%; women: 33.5%) [6].

Notably, only one national-level study investigated the association between obesity [body mass index (BMI) ≥ 30] and some noncommunicable diseases such as type 2 diabetes mellitus (T2DM) [odds ratio (OR) = 1.46; 95% confidence interval (CI): 1.12–1.91], hypercholesterolemia (OR = 1.57; 95% CI: 1.16–2.14), and hypertension (OR = 3.63; 95% CI: 2.70–4.88) [7].

The association between obesity and cancer is still unclear, although obesity might influence cancer outcomes [8]. However, obesity is associated with respiratory symptoms and lung diseases, including exertional dyspnea, obstructive sleep apnea syndrome, obesity hypoventilation syndrome, chronic obstructive pulmonary disease, and asthma [9].

The ‘obesity paradox’ shows no association with cardiovascular (CV) diseases or association with better survival and fewer CV events in certain groups of people, such as very elderly individuals or those with certain chronic diseases with elevated BMI [10]. However, obesity is strongly linked to metabolic abnormalities, which in turn linked to CV. Obesity is a risk factor for hypertension and hypercholesterolemia, and it plays an important role in the metabolic syndrome criteria [10]. Besides, obesity was significantly associated with developing rheumatoid arthritis (OR = 1.24, 95% CI: 1.01–1.53; adjusted for smoking status) [11].

However, no national study has investigated the association between obesity and cancer, lung diseases, CV diseases, sleep apnea, and rheumatoid arthritis.

Thus, the present study aimed to assess the prevalence of obesity and its distribution within (regions, age, and gender) Saudi Arabia. In, addition, this study explored the current association between obesity and various health conditions.

## 2. Method

### 2.1. Study Design

This study is a nationwide cross-sectional survey conducted as phone interviews in June 2020.

### 2.2. Sampling and Sample Size

A proportional quota sampling technique was employed to acquire an equal distribution of participants stratified by age and gender across the 13 regions of Saudi Arabia. Two age groups based on the median age of Saudi Arabian adults (36 years) were used, leading to a quota of 52. The QPlatform^®^ data collection system, which had integrated eligibility and sampling modules, was used to control the sample distribution [12]. The eligibility module included three questions to determine the completeness of the sampling quota, including age, gender, and region. The sample size was calculated based on a medium effect size of approximately 0.25, with 80% power and 95% CI, to compare the age and gender across regions [13]. Thus, each quota required 90 participants, and the total targeted sample was 4680 participants. As the data collection system close the quota only after achieving the targeted sample, and as there are a group of phone call attempts happening simultaneously, on some occasions more than one participant can pass the eligibility process and there may be a sample increase in some of the quota above the targeted sample. Thus, a slightly larger sample size may be recruited.

### 2.3. Participant Recruitment

Participant recruitment was limited to Arabic-speaking Saudi residents who were ≥18 years old. A random phone number list was generated from the Sharik Association for Health Research to identify potential participants [14]. The Sharik database is composed of individuals who are interested in participating in future research projects and contains a growing number of registered participants that have reached more than 63,000 distributed across the 13 regions of Saudi Arabia [14]. Participants were contacted by phone on up to three occasions. If they did not respond, a new number with similar demographics is then generated from the database until the quota is completed and closed automatically. After obtaining consent to participate, the interviewer assessed the eligibility, based on the above-mentioned quota completion criteria.

### 2.4. Survey and Outcome Measures

The survey questions were adopted from the 2016–2017 National Health Interview Survey (NHANES) [15]. Questions included demographic information (age, gender, and region), diagnosed intermediate-risk factors (high blood pressure, and high cholesterol), obesity (measured as BMI using self-reported current height and weight in their last measurement), and diagnosed on-treatment chronic conditions (T2DM, CV diseases, cancer, lung diseases, liver diseases, colon diseases, sleep apnea, rheumatoid arthritis, thyroid disorders, peptic ulcer, and depression).

We used the Center for Disease Control and Prevention (CDC) BMI category status, which specifies the BMI below 18.5 kg/m^2^ as underweight, from 18.5 to 24.9 kg/m^2^ as normo-weight, from 25 to 29.9 kg/m^2^ as overweight, and 30 kg/m^2^ and above as obese [15].

Linguistic validation was performed to ensure that the translation from English to Arabic had the same intention as questions from the source questionnaire. The standard backward and forward translation was done. Two nutritionists and one research professional independently conducted the forward translation, and the backward translation was done separately by two professional translators. A focus group of seven participants was asked to discuss and answer the survey questions, and an updated version was tested again with another focus group. Afterward, the electronic version of the survey developed on the QPlatform, and a pilot test with 30 participants was interviewed by phone to ensure the accuracy, quality, and data integrity of the survey. Per the pilot study results and feedback from the researchers and interviewers, the questionnaire was edited further, and an improved version was developed. All questions had to be answered for the responses to be successfully submitted to the database. All data were coded and stored on the QPlatform database [12].

### 2.5. Primary Outcomes of Interest

Assessing the prevalence of obesity and its distribution within regions, age, and gender strata in Saudi Arabia.Explore the current association between obesity and various non-communicable diseases.

### 2.6. Ethical Considerations

The ethics committee of Sharik Association for Health Research approved this research project (Approval no.2020-3), according to the national research ethics regulations. Consent to participate were obtained verbally during the phone-interview with the participants and recorded on the data collection system.

### 2.7. Data Analysis

The prevalence of obesity was calculated using the frequency and percent, and the weighted prevalence was also calculated based on region population per the latest population census data by the General Authority of Statistics 2017 Census Report [16]. Multivariate logistic regression analyses were used to investigate the association between obesity and diseases adjusted to age and gender. Results were presented as OR and 95% CI. A *p*-value of <0.05 was used to indicate statistical significance. Data management and analyses were carried out using the Statistical Package for Social Sciences (SPSS, Armonk, NY, USA).

## 3. Results

### 3.1. Demographics & Response Rate

Of the 6239 participants contacted, 4709 participants responded and completed the interview with a response rate of 75.48%, across the 13 administrative regions of Saudi Arabia. Of them, 50.1% were female, and the mean age was 36.4 ± 13.5 [Range:18 to 90], and the median age was 36. Table 1 shows the demographic characteristics of the participants.

### 3.2. Obesity Prevalence and Distribution

The national weighted prevalence of obesity (BMI ≥ 30) was 24.7%, and the prevalence in the sample (unweighted) was 21.7%. Table 2 shows the prevalence of obesity by region, age group, and gender in this study sample.

### 3.3. Current Status of the Association between Obesity and Health Conditions

Obesity was significantly associated with T2DM, hypercholesterolemia, hypertension, lung diseases, rheumatoid arthritis, sleep apnea, colon diseases, and thyroid disorders. However, obesity was not significantly associated with CV diseases, cancer, diagnosed depression, liver diseases, and peptic ulcer. Table 3 presents the association between obesity and various chronic conditions.

## 4. Discussion

This study investigated the prevalence of obesity in Saudi Arabia from a recent national level survey conducted by phone-interviews. The national weighted prevalence of obesity (BMI ≥ 30) was 24.7% in this study. This weighted prevalence of obesity is decreasing compared to 25.6% in 2018 and 28.7% in 2013 [5,6]. This finding is also lower than other Middle Eastern countries such as the UAE and Kuwait [3,4].

There were no data to justify the decreasing trend in obesity in Saudi Arabia. However, there are many new regulations that happened in the last decade that may contribute to a future decrease in obesity. Some policy changes to promote a healthier lifestyle in Saudi Arabia may have a significant long-term effect for reducing obesity. Some of these keystone changes that happened between 2017 to 2020 were: passing the law allowing female fitness centers to open in Saudi Arabia by 2017, and more fitness centers for both genders are currently opening across the country [17]; the introduction of physical activity classes in female schools in 2017 [18]; the government initiated quality of life programs as part of the Vision 2030 plan, which include programs and resources to encourage people’s participation in exercise as well as healthy lifestyles [19]; the law of printing a meal’s calories on restaurant menus at the beginning of 2019, which was followed by large awareness increases in the Saudi community [20]; the introduction of an excise tax of 50% on sugar sweetened beverages and carbonated drinks, and 100% on Energy drinks in early 2019 [21]. Although the excise tax was unlikely to have had a significant effect in the short term on the whole population, it has proven elsewhere to have a dramatic impact on individuals that rely heavily on fizzy drink consumption for their daily caloric intake [22,23].

A 2005 national survey categorized the obesity prevalence among different regions of the country, with the highest two regions being Hail (33.9%) and the Eastern Region (27.7%), while the lowest two regions were Madinah (15.1%) and Jazan (11.7%) [24]. In our study, the highest obesity region was the Eastern Region (29.4%), followed by Riyadh (26.9%), while the lowest was Baha (14.0%), then Asir (18.0%). It is unclear why there are differences between the regions, however, they may be related to specific regional factors such as food related cultures, or differences in lifestyles and behaviors related to obesity that may need future investigations. These differences are also an opportunity to find and apply local strategies to reduce obesity in other high prevalence regions.

In terms of describing the current status of obesity association with health conditions, this study investigated the associations between obesity and various diagnosed conditions. This study found a significant association between obesity and diagnosed conditions, including T2DM, hypercholesterolemia, hypertension, lung diseases, rheumatoid arthritis, sleep apnea, colon diseases, and thyroid disorders. The findings were in agreement with global literature and confirmed the association of obesity with these conditions regardless of the sociodemographic differences between the countries [8,10,11]. In addition, these findings highlighted the current co-existence of obesity and chronic health conditions. Such a co-existence may increase the burden of disease on individuals and lower their disease management outcomes. Special focus on these groups is important for clinical practices in Saudi Arabia. 

One of the limitations in this study is that weight and height were self-reported. Self-reporting may lead to overestimating height and underestimating weight in some demographical groups [25]. Also, this study was limited to an analysis of cross-sectional data and a lack of temporality, which prevented cause-effect relationship investigation between obesity and health conditions. Quota sampling, instead of random sampling, was used for participant recruitment, which might have a risk of selection bias. The use of a research participant database might also introduce some bias. However, currently in Saudi Arabia the only way to generate a random national level sample is via household surveys, which also have some significant limitations due to sociocultural factors and were not possible to conduct when COVID-19 restrictions were in place. The recruitment and sampling methods used in this research project were used successfully in various recent national projects in Saudi Arabia [26,27]. Also, quota sampling allowed for the recruitment of a balanced study sample in terms of gender and age. Data integrity checks, inherent to the QPlatform data collection system, minimized invalid or erroneous data entries. Linguistic validation and questionnaire piloting were employed to strengthen the questionnaire’s reliability. The study analyzed a large study sample with high representativeness of adults in Saudi Arabia. 

## 5. Conclusions

Although obesity shows a decreasing trend in Saudi Arabia, almost one-quarter of this study’s sample were obese. It is currently associated with many health conditions that can affect an individuals’ quality of life and create an economic burden on the country. This evidence highlights the need for action to focus more on obesity in Saudi Arabia.

## Figures and Tables

**Table 1 healthcare-09-00311-t001:** Demographic characteristics of participants.

Variable	Proportions *n* (%)		
	Female	Male	Total
Age Groups			
18–19	120 (47.1)	135 (52.9)	255 (5.4)
20–29	827 (53.1)	729 (46.9)	1556 (33.0)
30–39	469 (46.5)	540 (53.5)	1009 (21.4)
40–49	577 (55.3)	467 (44.7)	1044 (22.2)
50–59	260 (46.8)	295 (53.2)	555 (11.8)
60+	105 (36.2)	185 (63.8)	290 (6.2)
Sex			
Female	-	-	2358 (50.1)
Male	-	-	2351 (49.9)
Regions			
Jouf	180 (49.9)	181 (50.1)	361 (7.7)
Tabuk	182 (50.3)	180 (49.7)	362 (7.7)
Hail	179 (49.9)	180 (50.1)	359 (7.6)
Madinah	184 (50.3)	182 (49.7)	366 (7.8)
Qassim	181 (50.1)	180 (49.9)	361 (7.7)
Makkah	181 (50.0)	181 (50.0)	362 (7.7)
Riyadh	183 (50.3)	181 (49.7)	364 (7.7)
Eastern Region	180 (50.0)	180 (50.0)	360 (7.6)
Baha	181 (49.7)	183 (50.3)	364 (7.7)
Asir	184 (50.3)	182 (49.7)	366 (7.8)
Jazan	181 (50.1)	180 (49.9)	361 (7.7)
Najran	181 (50.0)	181 (50.0)	362 (7.7)
Northern Borders	181 (50.1)	180 (49.9)	361 (7.7)

**Table 2 healthcare-09-00311-t002:** Prevalence of obesity in the sample (BMI ≥ 30) prevalence by region, age group, and gender.

Region	Prevalence (*n*)
Asir	18.0% (66)
Baha	14.3% (52)
Eastern Region	29.4% (106)
Hail	20.1% (72)
Jazan	19.9% (72)
Jouf	26.6% (96)
Madinah	23.0% (84)
Makkah	25.4% (92)
Najran	20.2% (73)
Northern Borders	21.1% (76)
Qassim	18.3% (66)
Riyadh	26.9% (98)
Tabuk	19.3% (70)
Age Group (Years)	
18–19	14.1% (36)
20–29	14.8% (231)
30–39	18.1% (183)
40–49	29.8% (311)
50–59	32.8% (182)
60+	27.6% (80)
Gender	
Female	25.5% (601)
Male	17.9% (422)
Total	21.7% (1023)

**Table 3 healthcare-09-00311-t003:** Crude and adjusted odds ratio (OR) for the association between obesity and different chronic diseases.

Disease	Proportion (*n*)	Crude OR (95% CI) (*p*-Value)	Adjusted OR (95% CI) (*p*-Value)
Type 2 Diabetes			
Obese	10.2% (104)	2.13 (1.66–2.74) (<0.001)	1.52 (1.16–1.98) (0.002)
Non-obese	5.0% (186)	Reference	Reference
Hypercholesterolemia			
Obese	17.6% (180)	2.22 (1.83–2.71) (<0.001)	1.69 (1.36–2.09) (<0.001)
Non-obese	8.8% (323)	Reference	Reference
Hypertension			
Obese	22.7% (232)	2.05 (1.72–2.45) (<0.001)	1.61 (1.33–1.96) (<0.01)
Non-obese	12.5% (461)	Reference	Reference
Cardiovascular Diseases			
Obese	3.9% (40)	1.52 (1.05–2.22) (0.02)	1.24 (0.84–1.83) (0.28)
Non-obese	2.6% (96)	Reference	Reference
Lung Diseases			
Obese	6.0% (61)	1.73 (1.27–2.37) (0.001)	1.69 (1.23–2.34) (0.001)
Non-obese	3.5% (130)	Reference	Reference
Cancer			
Obese	1.4% (14)	2.31 (1.18–4.53) (0.015)	1.66 (0.84–3.30) (0.14)
Non-obese	0.6% (22)	Reference	Reference
Rheumatoid Arthritis			
Obese	10.2% (104)	2.45 (1.89–3.16) (<0.001)	1.57 (1.19–2.05) (0.001)
Non-obese	4.4% (163)	Reference	Reference
Diagnosed Depression			
Obese	3.7% (38)	1.30 (0.89–1.90) (0.17)	1.27 (0.86–1.88) (0.22)
Non-obese	2.9% (106)	Reference	Reference
Sleep Apnea			
Obese	2.3% (24)	2.04 (1.23–3.70) (0.006)	1.82 (1.08–3.04) (0.02)
Non-obese	1.2% (43)	Reference	Reference
Renal Diseases			
Obese	2.8% (29)	2.17 (1.36–3.45) (0.001)	1.82 (1.13–2.94) (0.01)
Non-obese	1.3% (49)	Reference	Reference
Liver Diseases			
Obese	1.1% (11)	1.20 (0.61–2.39) (0.59)	0.96 (0.48–1.93) (0.74)
Non-obese	0.9% (33)	Reference	Reference
Colon diseases			
Obese	15.3% (157)	1.54 (1.26–1.88) (<0.001)	1.31 (1.06–1.60) (0.01)
Non-obese	10.6% (389)	Reference	Reference
Thyroid Disorders			
Obese	7.6% (78)	2.37 (1.77–3.18) (<0.001)	1.80 (1.33–2.44) (<0.001)
Non-obese	3.4% (124)	Reference	Reference
Peptic Ulcer			
Obese	2.9% (30)	1.13 (0.75–1.71) (0.57)	0.93 (0.61–1.42) (0.73)
Non-obese	2.6% (96)	Reference	Reference

## Data Availability

Available from Sharik Association for Health Research upon request.

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
