# Peer review of "Obesity in Saudi Arabia in 2020: Prevalence, Distribution, and Its Current Association with Various Health Conditions"

_healthcare, 2021, doi:10.3390/healthcare9030311_

Round 1

Reviewer 1 Report

Review of the Manuscript healthcare-1130168

Title: Obesity in Saudi Arabia in 2020: Prevalence, Distribution, and Impact on Population Health

Dear Authors,

The research on the current prevalence of obesity in Saudi Arabia is interesting. However, there are several issues that the authors should clarify. Please reply to the following comments:

  • the lack of line numbering makes it difficult to precisely refer to comments.

Introduction

  • The aims of the study were: to assess the prevalence of obesity in Saudi Arabia and to explore the impact of obesity on health by investigating its association with other non-communicable diseases. However, all data are self-reported and it is difficult (or rather impossible) to establish cause and effect relationships on the basis of such data. Authors can only determine the status quo of the population that responded by telephone.

Methods

  • 5. Primary Outcomes point 2. As mentioned above exploring the impact of obesity on health by investigating the association between obesity and non-communicable diseases is not possible on the basis of this interview, as the authors assess the current health status of the population.

Discussion

  • The authors cannot draw strong conclusions about the impact of obesity on population health from observational data (self-reported). This is an oversimplification and may lead biased conclusions.

To summarize, the article requires minor revisions before being published.

Author Response

=====================

Reviewer #1:

The research on the current prevalence of obesity in Saudi Arabia is interesting. However, there are several issues that the authors should clarify.

Authors’ Response: Noted with many thanks for reviewing this work.

1- The lack of line numbering makes it difficult to precisely refer to comments

Authors’ Response: Sorry for the inconvenience, The line numbering is now included.   

2- The aims of the study were: to assess the prevalence of obesity in Saudi Arabia and to explore the impact of obesity on health by investigating its association with other non-communicable diseases. However, all data are self-reported and it is difficult (or rather impossible) to establish cause and effect relationships on the basis of such data. Authors can only determine the status quo of the population that responded by telephone

Authors’ Response: We totally agree with the reviewer. This study was not aimed to established cause effect relationship instead it was aimed to described currents status of association between obesity and health conditions diseases. In addition, we clarify that this point in the limitation section and in various areas of the manuscript to make this point clear for the readers.

3- Primary Outcomes point 2. As mentioned above exploring the impact of obesity on health by investigating the association between obesity and non-communicable diseases is not possible on the basis of this interview, as the authors assess the current health status of the population.:

Authors’ Response: Agree, Primary Outcomes was modified to clarify this point.

4- The authors cannot draw strong conclusions about the impact of obesity on population health from observational data (self-reported). This is an oversimplification and may lead biased conclusions.

Authors’ Response: Agree, the conclusion has been revised to address this point.

Reviewer 2 Report

Review of the article Obesity in Saudi Arabia in 2020: Prevalence, Distribution, and Impact on Population Health.

Thank you for inviting me to the review. An interesting scientific report that says, among others about obesity in Saudi Arabia.

Abstract

The authors write ... "Although obesity shows a decreasing trend, it is still almost affecting one-quarter of the population ..." the number of 4,709 surveyed people is rather not the country's population but a modest portion - it is worth correcting the sentence.

Admission The authors write ...: Per the 2016 World Health Organization data, about 13% of the world's adult population (male: 11%; female: 15%) were obese [1] ... "these are data from 2016 - can you find any scientific reports from 2018, 2019 or 2020? - The data may show slightly different values. In Saudi Arabia, have any other groups been tested for obesity in the past 20 years? - if so, it is worth adding. There is clearly a lack of basic information on the state of obesity in other countries, e.g. directly abroad? Eg Yemen, Oman, Iraq, Israel ...

Methodology The authors write… ”total targeted sample was 4680 participants…” in turn further that the research group is 4709? The authors write ... "Participants were contacted by phone on up to three occasions .." I believe that the information obtained over the phone carries with it a lot of overestimation in the responses to the height and weight of the body. Hence, these values ​​can be definitely different. - however, the main issue in such studies is direct measurements on the scale, e.g. Tanita 360 and Seca height meter ... Additionally, there is no information in the methodology as to whether people were informed about how to measure their weight and weight! (with or without clothes, upright, without shoes, etc ...).

Results The authors write ... "Of them, 50.1% were female, and the mean age was 36.4 ± 13.5 [18.90] ..." where did these footnotes 18 and 90 come from ??? Table 1 lacks a breakdown by gender in age groups and regions. This will give a better picture, for example, for comparisons with previous studies in these areas. Discussion The authors write… ”first, passing the law of allowing female fitness centers to open in Saudi Arabia… ”- if there were no such questions for the respondents, eg whether you attend gyms - no such conclusions can be drawn, because they are not related to the results of the research. The authors write ... "Second, in 2017 physical activity classes were introduced in female school ..." - young people rather go to school, and these physical activity classes began to be introduced only in 2017 - so it may only affect a small female group at the age of 19 -22. - I think it has a little relationship now, but it will be very important in the future. The authors write ... "In addition, the government-initiated quality of life programs as part of Vision 2030 plan and which include programs and resources to encourage people participation in exercise as well healthy life style ..." - this is a program that recently appeared in Arabia Saudi Arabia, therefore, I think that at the moment it is not very related to the results of the research. There are no comparisons to other studies from other countries in the discussion. There is also a lack of basic information, e.g. on (diet, physical activity, and smoking), and the authors write about it in the methodology. Conclusions It cannot be concluded that…: in Saudi Arabia, it is still almost affecting one-quarter of the population… ”- this is not a representative group.

References There are few publications from different countries!

Author Response

====================================

Reviewer #2:

Review of the article Obesity in Saudi Arabia in 2020: Prevalence, Distribution, and Impact on Population Health.Thank you for inviting me to the review. An interesting scientific report that says, among others about obesity in Saudi Arabia:

Authors’ Response: Noted with many thanks for reviewing this work.

1- The authors write ... "Although obesity shows a decreasing trend, it is still almost affecting one-quarter of the population ..." the number of 4,709 surveyed people is rather not the country's population but a modest portion - it is worth correcting the sentence..

Authors’ Response:. Agree, the sentence in both abstract and conclusion were revised.

2- Admission The authors write ...: Per the 2016 World Health Organization data, about 13% of the world's adult population (male: 11%; female: 15%) were obese [1] ... "these are data from 2016 - can you find any scientific reports from 2018, 2019 or 2020?

Authors’ Response:. Agree with the reviewer. However, WHO have published this report on April -2020, but it shows statistics for 2016 estimating the prevalence of overweight and obesity. No updated global statistics on obesity by WHO were founded.

3- The data may show slightly different values. In Saudi Arabia, have any other groups been tested for obesity in the past 20 years? - if so, it is worth adding. There is clearly a lack of basic information on the state of obesity in other countries, e.g. directly abroad? Eg Yemen, Oman, Iraq, Israel...

Authors’ Response:. Agree, there were 2 national surveys in 2000 and 2013 that was reported in Saudi Arabia. We also included some regional obesity data to the introduction.

4- Results The authors write ... "Of them, 50.1% were female, and the mean age was 36.4 ± 13.5 [18.90] ..." where did these footnotes 18 and 90 come from ???

Authors’ Response: Thanks for highlighting this typo, actually it is the age [range: 18 to 90], corrected now.

5-  Table 1 lacks a breakdown by gender in age groups and regions. This will give a better picture, for example, for comparisons with previous studies in these areas.

Authors’ Response: Agree, Gender cross-tabulation was included in table 1.

6- The authors write… ”first, passing the law of allowing female fitness centers to open in Saudi Arabia… ”- if there were no such questions for the respondents, eg whether you attend gyms - no such conclusions can be drawn, because they are not related to the results of the research.  The authors write ... "Second, in 2017 physical activity classes were introduced in female school ..." - young people rather go to school, and these physical activity classes began to be introduced only in 2017 - so it may only affect a small female group at the age of 19 -22. - I think it has a little relationship now, but it will be very important in the future.

Authors’ Response: Agree. We clarified that this collection of policies related to healthier life style may play an important role in reducing obesity in the future.

8- The authors write ... "In addition, the government-initiated quality of life programs as part of Vision 2030 plan and which include programs and resources to encourage people participation in exercise as well healthy life style ..." - this is a program that recently appeared in Arabia Saudi Arabia, therefore, I think that at the moment it is not very related to the results of the research.

Authors’ Response: Agree. We clarified that this collection of policies related to healthier life style may play an important role in reducing obesity in the future.

9- There are no comparisons to other studies from other countries in the discussion.

Authors’ Response: Done. The discussion has been updated to highlight comparison with some middle eastern countries such as UAE and, Kuwait.

There is also a lack of basic information, e.g. on (diet, physical activity, and smoking), and the authors write about it in the methodology.

Authors’ Response: Agree. This part was removed because it is irrelevant to the focus and scope of the study.

10- Conclusions It cannot be concluded that…: in Saudi Arabia, it is still almost affecting one-quarter of the population… ”- this is not a representative group.

Authors’ Response: Agree, the conclusion have been revised to address this point.

Thank you for your valuable feedback and comments.

Reviewer 3 Report

AIms of this paper are to assess the prevalence of obesity and its distribution within regions, age, and gender in Saudi Arabia. The authors also  investigated the association between obesity and non-communicable diseases.

Although the work was carried out in a formally correct manner, there are a number of criticisms that should be addressed.

  1. The authors should underline that the obesity paradox is a medical hypothesis that occurs only in certain groups of people, such as very elderly individuals or those with certain chronic diseases.
  2. Is it reliable to ask for weight and height by telephone? What are the experiences in the past with this type of measurement? Have there been any cross-checks of any kind?
  3. The authors should explain why there are regional differences in the prevalence of obesity.
  4. Some of the data collected could be very useful to make the work more interesting. For example behavioral risk factors (diet, physical activity, and smoking) were not considered. In a study that aims to detect risk factors these data are very important and should be presented and discussed.
  5. The discussion only comments on the data in a superficial way. It does not offer any particular input for further work or implications for clinical practice.

Author Response

=====================================

Reviewer #3:

Aims of this paper are to assess the prevalence of obesity and its distribution within regions, age, and gender in Saudi Arabia. The authors also  investigated the association between obesity and non-communicable diseases. Although the work was carried out in a formally correct manner, there are a number of criticisms that should be addressed.

Authors’ Response: Noted with many thanks for reviewing this work the valuable feedback which contributed to improving this article.

1- The authors should underline that the obesity paradox is a medical hypothesis that occurs only in certain groups of people, such as very elderly individuals or those with certain chronic diseases.

Authors’ Response: Agree. We revised the sentence to clarify this point.

2- Is it reliable to ask for weight and height by telephone? What are the experiences in the past with this type of measurement? Have there been any cross-checks of any kind?

Authors’ Response: This is an important point to address. A review of more than 60 studies comparing self-reported weight and height showed tendency for underestimation of overweight (from 1.8%-points to 9.8%-points) and obesity (from 0.7%-points to 13.4%-points) prevalence by self-report. We addressed this point in the limitation section.

3- The authors should explain why there are regional differences in the prevalence of obesity.

Authors’ Response: As this is one of the first studies in a long time to zoom in to regional prevalence, it is unclear for the author why such differences is happening especially with the lack of related literature. We included some assumptions that may be relevant to future investigation.

4- Some of the data collected could be very useful to make the work more interesting. For example behavioral risk factors (diet, physical activity, and smoking) were not considered. In a study that aims to detect risk factors these data are very important and should be presented and discussed.

Authors’ Response: We totally agree with the reviewer that behavioral risk factors are important. However, during the drafting of the manuscript we noticed that it is out of this paper scope and adding it would require major focus on its prevalence, association with obesity, and demographics which will expand the article beyond acceptable wording limit and may overwhelm the reader. However, we are considering future analysis and investigation in this area. 

5- The discussion only comments on the data in a superficial way. It does not offer any particular input for further work or implications for clinical practice

Authors’ Response: Thank you, the discussion has been revised to include more depth.

Thank you for your valuable feedback and comments.

Round 2

Reviewer 2 Report

Review of the article Obesity in Saudi Arabia in 2020: .....

Round 2

Abstract - ok

Admission - there is still a lack of basic information on the state of obesity in other countries, e.g. directly abroad? Eg Yemen, Oman, Iraq, Israel ...

Methodology - still no answer? - The authors write… ”total targeted sample was 4680 participants…” in turn further that the research group is 4709?

- still no answer? - there is no information in the methodology as to whether people were informed about how to measure their weight and weight! (with or without clothes, upright, without shoes, etc…).

Results - approx

Discussion - still no answer? - ... "first, passing the law of allowing female fitness centers to open in Saudi Arabia ..." - if there were no such questions to the respondents, for example, do you attend gyms - no such conclusions can be drawn, because they are not related to the test results.

References The newly introduced publications are technically incorrect (no. 2,3,4,26,27) 

Author Response

Reviewer #2:

Abstract - ok

Admission - there is still a lack of basic information on the state of obesity in other countries, e.g. directly abroad? Eg Yemen, Oman, Iraq, Israel ...

Authors’ Response: The requested information is now included in the introduction section.

Methodology - still no answer? - The authors write… ”total targeted sample was 4680 participants…” in turn further that the research group is 4709?

Authors’ Response: As the data collection system close the quota only after achieving the targeted sample, and as there are a group of phone call attempts happening simultaneously, on some occasions more than one participant can pass the eligibility process and there for there may be some sample increase in some quota above the targeted sample. Thus, a slightly larger sample size may be recruited. This point is now clarified in the sampling section.

- still no answer? - there is no information in the methodology as to whether people were informed about how to measure their weight and weight! (with or without clothes, upright, without shoes, etc…).

Authors’ Response: As this is a live phone call interview, we asked the participants about their current weight and height from their last measurement but no further questions of how these measurements were obtained. We clarified this point further in the method section.

Discussion - still no answer? - ... "first, passing the law of allowing female fitness centers to open in Saudi Arabia ..." - if there were no such questions to the respondents, for example, do you attend gyms - no such conclusions can be drawn, because they are not related to the test results.

Authors’ Response: There were no questions related to these policies, we removed any suggestion that these policies may be related to the results. However, they are significant changes relevant to the readers to understand the current policies that may impact the future of obesity in the country.

References The newly introduced publications are technically incorrect (no. 2,3,4,26,27)

Authors’ Response: We formatted the references to the required style.

Thank you for your valuable feedback and comments.

Reviewer 3 Report

The authors have responded to all my comments. The paper can now be published.

Author Response

Reviewer #3:

The authors have responded to all my comments. The paper can now be published.

Authors’ Response: Noted. Thank you for your valuable feedback and comments.